# LcSAO1, an Unconventional DOXB Clade 2OGD Enzyme from *Ligusticum chuanxiong* Catalyzes the Biosynthesis of Plant-Derived Natural Medicine Butylphthalide

**DOI:** 10.3390/ijms242417417

**Published:** 2023-12-13

**Authors:** Xueqing Chen, Xiaopeng Zhang, Wenkai Sun, Zhuangwei Hou, Bao Nie, Fengjiao Wang, Song Yang, Shourui Feng, Wei Li, Li Wang

**Affiliations:** 1Shenzhen Branch, Guangdong Laboratory of Lingnan Modern Agriculture, Key Laboratory of Synthetic Biology, Ministry of Agriculture and Rural Affairs, Agricultural Genomics Institute at Shenzhen, Chinese Academy of Agricultural Sciences, Shenzhen 518000, Chinaalfredhou8@gmail.com (Z.H.);; 2State Key Laboratory of Biocontrol and Guangdong Provincial Key Laboratory of Plant Resources, School of Life Sciences, Sun Yat-sen University, Guangzhou 510275, China; Tars9612@outlook.com

**Keywords:** butylphthalide, desaturation, Fe (II)/2-oxoglutarate-dependent dioxygenase, *Ligusticum chuanxiong*

## Abstract

Butylphthalide, a prescription medicine recognized for its efficacy in treating ischemic strokes approved by the State Food and Drug Administration of China in 2005, is sourced from the traditional botanical remedy *Ligusticum chuanxiong*. While chemical synthesis offers a viable route, limitations in the production of isomeric variants with compromised bioactivity necessitate alternative strategies. Addressing this issue, biosynthesis offers a promising solution. However, the intricate in vivo pathway for butylphthalide biosynthesis remains elusive. In this study, we examined the distribution of butylphthalide across various tissues of *L. chuanxiong* and found a significant accumulation in the rhizome. By searching transcriptome data from different tissues of *L. chuanxiong*, we identified four rhizome-specific genes annotated as 2-oxoglutarate-dependent dioxygenase (2-OGDs) that emerged as promising candidates involved in butylphthalide biosynthesis. Among them, LcSAO1 demonstrates the ability to catalyze the desaturation of senkyunolide A at the C-4 and C-5 positions, yielding the production of butylphthalide. Experimental validation through transient expression assays in *Nicotiana benthamiana* corroborates this transformative enzymatic activity. Notably, phylogenetic analysis of LcSAO1 revealed that it belongs to the DOXB clade, which typically encompasses genes with hydroxylation activity, rather than desaturation. Further structure modelling and site-directed mutagenesis highlighted the critical roles of three amino acid residues, T98, S176, and T178, in substrate binding and enzyme activity. By unraveling the intricacies of the senkyunolide A desaturase, the penultimate step in the butylphthalide biosynthesis cascade, our findings illuminate novel avenues for advancing synthetic biology research in the realm of medicinal natural products.

## 1. Introduction

*Ligusticum chuanxiong* Hort., a perennial herbaceous plant in the Apiaceae family, has garnered considerable attention due to its significant medicinal properties. Its dried rhizome, known as chuanxiong (CX) in traditional Chinese medicine (TCM), is widely used to promote blood circulation and eliminate stasis. Additionally, the leaves of *L. chuanxiong* are used as a vegetable in southwestern China and South Korea and are believed to alleviate dizziness [1]. Studies have indicated that CX exhibits various effects, including the promotion of vasodilation [2,3], enhancement of cerebral circulation [4,5], inhibition of platelet aggregation [6,7], prevention of hepatic lipid accumulation [8], and demonstration of antioxidative activity [9,10]. Both in vitro and in vivo experiments have established the benefits of the cardio-cerebrovascular, nervous, and respiratory systems [11,12]. The therapeutic effects of CX are attributed to its intricate chemical composition. A comprehensive analysis has identified more than 263 chemical components from different parts of *L. chuanxiong*, and these constituents encompass phthalides, terpenes and their enols, phenolic acids, alkaloids polysaccharides, and organic acids and their esters [1].

Notably, phthalides stand out as pivotal active compounds within the volatile oil of plants from the Apiaceae family, such as *Angelica sinensis* and *L. chuanxiong* [3,13]. Phthalides have been found to possess a range of pharmacological activities, including anti-inflammatory [14,15], antioxidant, neuroprotective, and cardiovascular protective effects [16]. The characteristic aroma of CX might be attributed to phthalides, which are benzenoids generated from aromatic amino acids [17]. Notably, a significant constituent of these phthalides, butylphthalide (formula C_12_H_14_O_2_, Figure 1) contributes to the aromatic odor of CX oil [11]. This compound is well-known for its neuroprotective properties and was approved to treat the ischemic strokes by the State Food and Drug Administration of China in 2005 [18]. The levels of butylphthalide in *L. chuanxiong*, mainly obtained from the leaves and rhizome of *L. chuanxiong*, varied among samples collected from different geographic locations [19,20]. Chemical synthesis of butylphthalide is available but the antithrombotic and antiplatelet activity of the product was lower than in the butylphthalide of natural origin [21]. And limited availability poses a barrier to the medicinal use of butylphthalide. Synthetic biology has demonstrated its effectiveness in the production of natural products, making it a promising approach to enhance butylphthalide production.

The precise biochemical mechanisms underpinning butylphthalide biosynthesis remain enigmatic (Figure 1). It is likely that one polyketide synthase uses acetyl-CoA as a starter to form the scaffold [22], then tailoring enzymes are involved in the following steps [23]. Interestingly, research has shown that senkyunolide A can be transformed into butylphthalide at room temperature [24]. And four Fe (II)/2-oxoglutarate-dependent dioxygenase superfamily (2OGDs) enzymes divided into DOXC family clade have been identified in converting senkyunolide A to butylphthalide in CX [25]. We also found some candidate genes belong to DOXB. Generally, the proteins in the DOXB clade have a subtype of the 2OGD domain (the proyl 4-hydroxylase (P4Hs)) [26]. The function of P4Hs is involved in synthesis of the cell wall in plants and algae [27,28] and P4Hs hydroxylated proline residues of those cell wall proteins. While DOXB enzymes usually perform hydroxylation rather than desaturation, analogous desaturation reactions catalyzed by DOXB clade 2OGDs have been observed, even in *E. coli* [29].

Here, we collected the leaf, stem, and rhizome of *L. chuanxiong* for multi-omics analysis. Based on transcriptome and metabolome correlation analysis of these tissues and phylogenetic investigations, we identified and characterized the potential candidate genes involved in the pathway of butylphthalide biosynthesis, which is largely unknown. Additionally, we investigated the key active sites of the newly discovered enzyme. These investigations offer valuable insights into the intricate metabolic network of butylphthalide.

## 2. Results

### 2.1. Butylphthalide Measurement in Different Tissues of L. chuanxiong

To investigate the conversion, we measured the content of butylphthalide in different tissues of *L. chuanxiong* (Figure 2A, Appendix A) through ultra-performance liquid chromatography mass spectrometry (UPLC-MS). This analysis showed that butylphthalide is largely produced in the rhizome of *L. chuanxiong*. Besides butylphthalide, senkyunolide A was also detected in *L. chuanxiong* by our previous research. Senkyunolide A exhibited a distribution pattern in test tissues that closely resembled that of butylphthalide [25].

### 2.2. Transcriptome Analysis of L. chuanxiong

We first generated de novo assembled transcriptome from different tissues of *L. chuanxiong*, including the leaf, stem, and rhizome. A total of 201,743,065 raw RNA-Seq reads were generated from the transcriptome of the leaf, stem, and rhizome of *L. chuanxiong* (Appendix A). After removing adaptors and low-quality reads, a mean of 22,415,896 clean reads was obtained and subsequently used for de novo assembly. In total, 392,621 trinity contigs (N50 = 1951 bp) and 197,194 unigenes were generated. The length of the assembled transcripts ranged from 201 bp to 197,194 bp with N50 of 1166 bp (Appendix A). Benchmarking Universal Single-Copy Orthologs (BUSCO) assessment revealed that 1402 out of 1614 (86.9%) core Embryophyta genes were covered by the assembly, indicating good assembly quality. A total of 91,968 unigenes (44.64% of all unigenes) were annotated, among which 88,911 and 53,395 unigenes had at least one high homology hit in the Nr database and SwissProt database, respectively.

A total of 40,062 unigenes were enriched into three functional GO categories: biological process (BP), cellular component (CC), and molecular function (MF). Most unigenes were annotated in the CC category. In the CC category, these unigenes were clustered into 9928 GO terms, among which “Plasma membrane” was the largest subcategory. For BP and MF categories, “Protein phosphorylation” and “Protein binding” were the largest subcategories, respectively. Additionally, a total of 34,135 unigenes were assigned in the KEGG database (Appendix A), which showed that most unigenes enriched in “mitogen-activated protein_kinase”, “cyclin-dependent kinase”, and “disease resistance protein”. Pfam annotation also showed that genes containing “Protein kinase domain” were the most commonly identified in the *L. chuanxiong* transcriptome.

### 2.3. Screening of Candidate Genes for Butylphthalide

In order to screen candidate genes participating in the catalysis of senkyunolide A into butylphthalide, we identified the differentially expressed genes (DEG) for two contrast groups (rhizome vs. leaf, rhizome vs. stem) with the default settings in DESeq. We found 19,197 and 4396 root up-regulated genes, respectively (Appendix A). The intersection of the two differentially expressed gene sets produced 3445 DEGs (Appendix A), which were considered to be rhizome up-regulated genes.

Considering the structural similarity between butylphthalide and senkyunolide A, we hypothesized that senkyunolide A could be converted into butylphthalide by desaturation, which could be conducted by 2OGDs. To identify the putative 2OGD genes in the rhizome up-regulated gene set, those genes annotated with PF13661 (characteristic functional domain of 2OGD) were extracted. As a result, only four 2OGD genes containing full-length ORFs and protein sequences longer than 300 amino acids were screened out (Figure 2B, Appendix A). As the expressional difference of LcSAO1 between rhizome and leaf was the greatest (>14-fold) among the four complete 2OGDs, we selected it for the following experimental verification.

### 2.4. LcSAO1 Converts Senkyunolide A to Butylphthalide

To identify the enzymatic activity of LcSAO1 (Figure 3A), the full-length coding region of LcSAO1 was cloned from cDNA prepared from the rhizome of *L. chuanxiong*. We expressed LcSAO1 into the pMal-c5x vector in *E. coli*, and the proteins were purified for the in vitro activity assay (Appendix A). Unfortunately, the enzyme reaction product was not detected by GC-MS (Appendix A). Although, we followed the protocol from previous literature for 2-OGD activity assay in vitro exactly [30,31], there are still a lot of factors that affect the enzymatic activity of 2-OGD in vitro and in vivo [32].

To confirm the enzymatic activity of LcSAO1, we utilized Agrobacterium-mediated transient expression to express LcSAO1 from the pSuper1300 vector in *N. benthamiana*. Following senkyunolide A treatment on the fourth day post-infiltration, *N. benthamiana* leaves were collected and ethyl acetate extracts were analyzed by GC-MS to evaluate the accumulation of butylphthalide. Compared with the empty vector control, the butylphthalide was detected in the leaves of *N. benthamiana* expressing LcSAO1 (Figure 3B,C). Taken together, we discovered a novel enzyme, LcSAO1, in the biological conversion of senkyunolide A into butylphthalide.

### 2.5. LcSAO1 Is a Desaturase That Unconventionally Belongs to the DOXB Family

The result of the blast analysis revealed that LcSAO1 had a high identity (63%) with the AtP4H7 (AT3G28480) (Appendix A and Appendix A), suggesting LcSAO1 was a member of the DOXB family of the 2OGD superfamily. The multiple sequence alignment conducted with four previously reported 2-OGD proteins (DOXC clade) revealed sequence identities ranging from 10.4167% to 12.9167% [25] (Appendix A). Furthermore, the predicted protein structure of LcSAO1 was found to be almost completely different from the four 2-OGDs proteins (Appendix A), indicating that LcSAO1 represents a distinct protein entity separate from the four 2-OGD proteins previously identified [25]. To explore the phylogenetic position of *LcSAO1*, a maximum likelihood (ML) phylogenetic tree of the DOXB family was constructed, which incorporated all DOXB genes of the other six Apiaceous species, *Arabidopsis thaliana*, and *Oryza sativa* (Figure 4, Appendix A). It revealed that LcSAO1 along with the other three DOXB genes of *L. chuanxiong* was grouped into the DOXB3 clade. LcSAO1 is closely clustered with three single-copy orthologous genes from other Apiaceous species (*Apium graveolens*, *Coriandrum sativum*, and *Bupleurum chinense*). Interestingly, *A. graveolens* and *C. sativum* had also been reported to contain butylphthalide, implying the potential function of orthologous *LcSAO1* genes in the butylphthalide accumulation in the three species. The absence of butylphthalide in *Bupleurum chinense*, an early divergent species in the Apioideae subfamily, may indicate the different functions of SAO in the early stage of Apiaceous evolution.

### 2.6. Key Catalytic Sites Identification Based on Structure Modelling and Activity Assay

In order to further understand the catalytic function of LcSAO1, we used the latest structure prediction tool AlphaFold2 to obtain the 3D structure models of LcSAO1. In addition to some disordered sequences at the N-terminal and C-terminal, most regions of the predicted 3D structure showed high accuracy with the pLDDT > 90, which represents the confidence level of a single amino acid residue. In a few parts of the sequence, pLDDT was greater than 70, but the protein skeleton was still maintained. The predicted structure with high accuracy can be used for the following analysis. The structure obtained by AlphaFold2 was then optimized using FastRelax module of multi-functional protein design platform Rosetta to eliminate irrational conformations in the structure [33]. The structure of LcSAO1 is highly similar to the three-dimensional structure of DOXB family proteins, whose crystal structure has been resolved [34], which implies its function of hydroxylation (Appendix A). Next, senkyunolide A was docked into the active site of LcSAO1 using RosettaScripts with Ligand_Docking, followed by conformation optimizing by FastRelax [33].

We analyzed the conserved information of LcSAO1 residues by the Position-Specific Scoring Matrix (PSSM) [35,36]. A higher score means more conservation. As we know, conserved residues generally have a function that is extremely important or irreplaceable for the function of the protein. We analyzed the active pocket (5 Å around substrate), chose the conserved residues, and mutated them to alanine based on the conserved information and structural model to explore the key sites that affect its catalysis (Figure 5). To further explore the potential active site residues of LcSAO1, we conducted site-directed mutagenesis using the list (Appendix A). To test the importance of those amino acid sites, we synthesized mutated LcSAO1 and subcloned into pSuper1300 vector. Then, the in vivo enzyme activity of LcSAO1 and its site-directed mutants was examined. Those genes were transiently expressed in the leaves of *N. benthamiana*. Enzymatic activity assays were performed by using senkyunolide A as the substrate. Among the 14 candidate amino acid residues, all of the 14 residues were possibly involved in the conversion of senkyunolide A into butylphthalide and all mutations decreased the enzymatic activity of LcSAO1. It is noteworthy that the substitutions of T98, S176, and T178 almost abolished enzymatic activity, to lower than 91%, 95.6%, and 94.3%, respectively (Figure 6, Appendix A).

## 3. Discussion

In plants, phthalides were mainly isolated from two genera *Ligusticum* and *Angelica* [13]. Due to the lack of relevant research, the biosynthesis pathway of phthalide is still unknown. Studies have shown that the biosynthesis pathway of phthalides may come from the acetate–malonate pathway, and the derivatives of ligustilide can be tailored through reduction, oxidation, decarboxylation, and dehydration [23]. CYP450s and the 2OGD superfamily were important structural tailor enzymes. The 2OGD enzymes facilitate a range of oxidative reactions, encompassing hydroxylation, desaturation, demethylation, and ring formation [37]. Although our previous studies identified four 2-OGDs (Lc2OGD1-4) belonging to the DOXC clade with the same desaturation activity with LcSAO1 [25], the results in this study demonstrated an unusual reaction catalyzed by an independently recruited DOXB clade *LcSAO1*, involving desaturation to produce an aromatic isobenzofuran.

It is believed that the 2-OGDs trace their origins back to a common ancestor predating the emergence of land plants [38]. Phylogenetic analysis showed that LcSAO1 belonged to the DOXB clade. The DOXB clade contains diverse 2-OGDs, which were mostly involved in proline hydroxylation. For example, prolyl 4-hydroxylases1 (P4H7) from *Arabidopsis thaliana* are key enzymes in modifying the post-translational formation of 4-hydroxyproline [27]. In *Arabidopsis thaliana*, P4H2, P4H5, and P4H13 play a role in extension hydroxylation, a process essential for maintaining proper cell wall architecture [39]. Surprisingly, LcSAO1 acts as a novel desaturation enzyme in the DOXB clade of 2-OGD.

Oxidation may contribute to the structural diversity of phthalides [23]. The functional role of 2-OGD in butylphthalide biosynthesis has been reported with a similar function in the Huperzine production from *P. tetrastichus* [31]. Specifically, the gene *Pt2OGD-3* from the DOXC clade showed its function in the formation of the aromatic pyridine ring by desaturation, which contributes to the diversity of the Lycopodium class of alkaloids. The mechanisms of desaturation by Fe/2OG enzymes remain poorly understood. Such transformation requires removing hydrogen atoms and perhaps requires two H.-accepting intermediates [40]. Despite the other functions of 2-OGD that have been reported [28], our study, along with the findings of Nett et al., revealed that 2-OGDs have desaturation functions. Though 2OGD-catalyzed desaturations are not common, the desaturation may promote stabilization from the adjacent heteroatom [31]. This suggests a possibly common pathway, by which different plants or microbes produce unique molecules via 2-OGD desaturation, as supported by substantial evidence [29,41,42,43].

The directed mutation of the activity sites of LcSAO1 (Figure 6) demonstrated a reduced or loss of function, which was also observed for the directed mutations of similar sites in P4H [44]. However, the function between LcSAO1 and P4H is different. In this study, we did not obtain the result from the recombinant protein in vitro enzyme assay when presenting Fe^2+^ with 2-OG (2-oxoglutarate) (Appendix A). Next, NADPH and FAD are further candidates as cofactors to confirm the activity of LcSAO1. Furthermore, the structure modelling of *LcSAO1* is crucial for articulating the key binding sites related to desaturation and helps explain the difference between LcSAO1 and P4H.

Another point worthy of mentioning is the conversion among the mono-phthalides. The structural diversity of mono-phthalides in *L. chuanxiong* has been identified [1]. Duric et al. demonstrated that ligustilide is the most stable one in the plant, and had a profound impact on the chemical conversion (convert to mono-, dimers-, and trimers phthalides) under light [45]. This means the structural diversity of phthalides may be the result of reduction, oxidation, cyclization, dehydration, and so on [23]. Despite the abovementioned exploration, the mechanism of interconversion between different phthalides, especially with regard to phthalide biosynthesis, remains ambiguous and warrants further investigation.

## 4. Materials and Methods

### 4.1. Plant Materials, Chemicals, and Reagents

*L. chuanxiong* (2n = 22) was collected from a cultivating farm in Pengzhou, Sichuan province, China. To obtain the *L. chuanxiong* transcriptome data, the rhizome, stem, and leaf of *L. chuanxiong* were immediately frozen in liquid nitrogen and stored at −80 °C until used for RNA isolation. The total RNA was isolated by the RNA plant reagent (Omega, Norcross, GA, USA) in accordance with the manufacturer’s protocol. Oligonucleotide primers were purchased from BGI (Beijing Genomics Institute, Shenzhen, China). The solvents for gas chromatography high-resolution mass spectrometry were from Sigma-Aldrich. The standard chemicals of butylphthalide and senkyunolide A were purchased from Chengdu Must Bio-technology Co., Ltd. (Chengdu, China).

### 4.2. RNA Extraction and Transcriptome Sequencing

RNA was extracted from three tissues (rhizome, stem, and leaf) using the Omega E.Z.N.A. plant RNA Kit (Omega, Norcross, GA, USA) according to the manufacturer’s protocol. Roughly 400 mg of each ground tissue was used for extraction. The RNA quality was validated using gel electrophoresis and the Qubit RNA IQ Assay (Thermo Fisher, CA, USA). Two micrograms of total RNA were utilized for stranded RNAseq library construction using the Illumina TruSeq stranded total RNA LT Sample Prep Kit (RS-122-2401 and RS-122-2402) (Illumina, San Diego, CA, USA). Multiplexed libraries were pooled and sequenced on an Illumina HiSeq4000 sequencer under the paired-end 150 nt mode. Three biological replicates were sequenced for each tissue.

### 4.3. De Novo Assembly and Functional Annotation of the Transcriptome of L. chuanxiong

The raw RNA-Seq reads were evaluated, filtered, and trimmed using FASTX-Toolkit to remove the low-quality reads and adaptors. After quality control, clean reads from three tissues were assembled into expressed sequence tag clusters (contigs) and de novo assembled into transcripts using Trinity 2.13.2 with default settings [46]. The longest transcript was chosen using Trinity in-house scripts. Transdecoder was used to predict coding regions [47]. The “embryophyte_odb10” database of Benchmarking Universal Single-Copy Orthologs (BUSCO) was used to evaluate the completeness of the assembly [48].

Gene ontology (GO), Kyoto Encyclopedia of Genes and Genomes (KEGG), and gene families were annotated based on GFAP default settings. Clean reads of each sample were aligned to the Trinity.fasta using the splice aware alignment program Hisat2 [49], and transcripts were identified using StringTie [50] with default parameters. PrepDE.py was used to calculate expression abundance. To identify the specifically root up-regulation genes, two contrast groups (rhizome vs. stem; leaf vs. rhizome) were statistically compared by the R package DESeq [51]. Differentially expressed genes (DEGs) were identified using the default method and settings. The candidate genes were selected from the intersection of rhizome up-regulated genes in the two contrast groups. GO enrichment and KEGG pathway enrichment analyses of the rhizome up-regulated genes were performed using R package ClusterProfiler [52]. The RNAseq datasets generated in the current study were deposited and are available at the National Genomics Data Center, Sequence Read Archive, under accession number CRA011442.

### 4.4. Determination of Butylphthalide Content in L. chuanxiong Tissues

Approximately 100 mg of plant tissues were dissected, transferred into grinding tubes, and then homogenized into fine powders on a ball mill using 5 mm stainless steel beads with shaking at 40 Hz for 30 s. Metabolites were extracted using 5 volumes (*w*/*v*) of 80% methanol at room temperature for 1 h. Extracts were centrifuged twice (13,000 rpm, 10 min) and supernatants were concentrated and collected for UPLC-MS analysis.

### 4.5. GC-MS Analysis of Metabolites

The sample preparation protocol for GC-MS was revised from previous studies [53,54]. The modified protocol was also used in our previous paper [25], and the detailed methods are described below. A sample volume of 1 µL was injected at a split ratio of 50:1 and subjected to analysis on an Agilent HP-5MS column (composed of 5% phenyl methyl siloxane, with dimensions of 30 m × 250 μm internal diameter and a film thickness of 0.25 μm). The initial column temperature was set at 50 °C for injection and then programmed to increase at a rate of 5 °C/min until reaching 250 °C. The spectrometers were operated in electron-impact (EI) mode, and metabolite analysis was conducted using both selected reaction monitoring (SRM) and full scan modes. The inlet temperature was maintained at 280 °C, while the ionization source temperature was held at 300 °C.

### 4.6. Heterologous Expression of Butylphthalide Precursor Genes in N. benthamiana

The described procedure was previously employed in our prior publication [25]. For the current study, LcSAO1 and its mutants (as listed in Appendix A) were inserted into the pSuper1300-GFP vector and introduced into GV3101 Agrobacterium electrocompetent cells following the manufacturer’s instructions. Individual strains were infiltrated into the abaxial side of 6-week-old *N. benthamiana* leaves using a needleless syringe. The entire leaf was carefully infiltrated with the Agrobacterium mixture. Three leaves, from three different plants, were infiltrated for each transient expression reaction. After 4–5 days of incubation, 1/3 of an infiltrated leaf was excised using scissors and placed in a 15 mL Safe-Lock tube (Eppendorf), snap frozen in liquid nitrogen, and lyophilized to dryness for subsequent metabolite extraction. To test for enzymatic activity on specific substrates absent in the plant (senkyunolide A), *Agrobacterium*-transformed leaves were co-infiltrated with a substrate solution (100 μM substrate in DI water) 3 days after *Agrobacterium* infiltration using a needleless syringe. This portion of the leaf was excised one day later, snap frozen in liquid nitrogen, and lyophilized to dryness for metabolite extraction.

### 4.7. Site-Directed Mutagenesis and Functional Characterization of Mutants

To conduct site-directed mutagenesis of *LcSAO1*, a two-round polymerase chain reaction (PCR) approach was employed. In the initial round of PCR reactions, the *pSuper1300-GFP/LcSAO1* plasmid was used as the template. We employed two pairs of primers, with each pair functioning as either the forward mutagenic primer or the reverse mutagenic primer (Appendix A). These primers were utilized to amplify two segments of *LcSAO1*, with overlapping ends. Subsequently, the purified PCR products were combined to create the full nucleotide sequences of mutants by second-round PCR reactions. Subsequently, the purified mutant DNA fragment was recombined into pSuper1300-GFP vectors after being digested with the XbaI and KpnI enzymes. All mutants were confirmed by sequencing performed by BGI (Beijing Genomics Institute, Shenzhen, China). The functional characterization of all mutants was performed as described above for the LcSAO1.

### 4.8. Heterologous Production, Purification, and In Vitro Reactions with LcSAO1

The recombinant plasmid *pMal-LcSAO1* was transformed into *E. coli*, and the strain was cultured in 1 mL liquid LB medium with 100 µg/mL Ampli at 37 °C, 220 rpm overnight. The cultured cells were resuspended in 10 mL liquid LB medium with 100 µM Ampli and cultured at 37 °C and 220 rpm until the OD_600_ reached 0.6. Subsequently, 0.5 mM IPTG was added, and the culture was continued for another 6 h at 37 °C. After induction, the cells were subjected to centrifugation at 8000 rpm for 6 min at 4 °C. The supernatant was discarded, and the cell pellet was resuspended in 20 mL lysis buffer (200 mM NaCl; 20 mM Tris-HCl pH 7.4; 1 mM EDTA, 1 mM PMSF). Resuspended cells were sonicated on ice for 8 s, and cooled down for 15 min. After another round of centrifugation, the supernatant was collected as a crude enzyme solution. The supernatant was loaded into an MBP Trap HP affinity column (GE Healthcare, Chicago, IL, USA) and eluted with lysis buffer containing 10 mmol/L maltose. The purity of MBP-LcSAO1 was detected by SDS-PAGE analysis and with Coomassie Blue staining. Fractions containing pure enzyme were combined and concentrated using Amicon^®^ Ultra-15 Centrifugal Filter Units (10 kDa molecular weight cut-off) via centrifugation at 4000 rpm for 20 min at 4 °C. Protein concentrations were measured via a 660 nm protein assay reagent assay kit. Purified protein solutions were then aliquoted, snap frozen with liquid nitrogen, and stored at −80 °C for future use. For in vitro assays with LcSAO1, reactions contained 50 μg/mL of purified LcSAO1, 25 μM of substrate (senkyunolide A), 10 mM 2-oxoglutarate, 0.5 mM FeSO_4_, and 10 mM ascorbate in a 200 μL reaction volume of potassium phosphate buffer [31], the pMal-c5x empty vector as a control. Reactions with each enzyme were incubated at 30 °C for 1 h, at which point they were quenched by diluting the reaction mixture 10-fold in ethyl acetate, and then analyzed by GC-MS.

### 4.9. Phylogenetic Analysis

A phylogenetic tree of *LcSAO1* orthologous was constructed using RAxML-NG (v1.1.0) [55] with 1000 bootstraps. The best fitting amino acid substitution model was selected as LG+I+G4+F by ModelTest-NG (v0.1.7) [56]. The protein alignment was obtained by MUCSLE (v5.1.linux64) [57]. All accession numbers used in this study are listed in Appendix A.

### 4.10. Homology Modelling, Ligand Docking, and Conservation Analysis

We obtained the predicted protein structure of LcSAO1 using AlphaFold2 (AF2) by modifying the script (http:github.com/deepmind/alphafold/blob/main/run_alphafold.py, accessed on 1 November 2023) on our own server [58]. Then, the conformation of protein models was optimized with the FastRelax module of Rosetta. To obtain the complex of LcSAO1 and senkyunolide A, it was docked using the Rosetta Ligand_docking in Rosetta Scripts with restriction of the distance between residues in the active center of LcSAO1 and senkyunolide A [33]. Amino acid conservation for LcSAO1 was determined through the utilization of PSIBLAST 2.2.27, which produced a Position Specific Scoring Matrix (PSSM). The PSSM was generated using the Uniref90 database, employing three iterations and applying an E-value cut-off of 0.01, as described in prior work [35,36].

### 4.11. Statistics and Reproducibility

Unless specifically stated otherwise, we conducted the statistical analysis using GraphPad Prism 9 for two-tailed Student’s *t*-test comparisons. The number of times each experiment was repeated is detailed in their respective sections.

## 5. Conclusions

In all, a new 2OGD (LcSAO1) belonging to DOXB was identified from *L. chuanxiong*. LcSAO1 is capable of catalyzing the desaturation of senkyunolide A at the C4-C5 position, leading to the production of butylphthalide. The structure predicted by AlphaFold2, along with docking, and site-directed mutagenesis characterization, unveiled important residues (T98, S176, and T178) in enzyme activity. This study will pave the way for the further characterization of DOXB clade 2OGDs and the expansion of the biosynthesis pathway of butylphthalide. They will also offer valuable insights into the intricate metabolic network of butylphthalide.

## Figures and Tables

**Figure 1 ijms-24-17417-f001:**
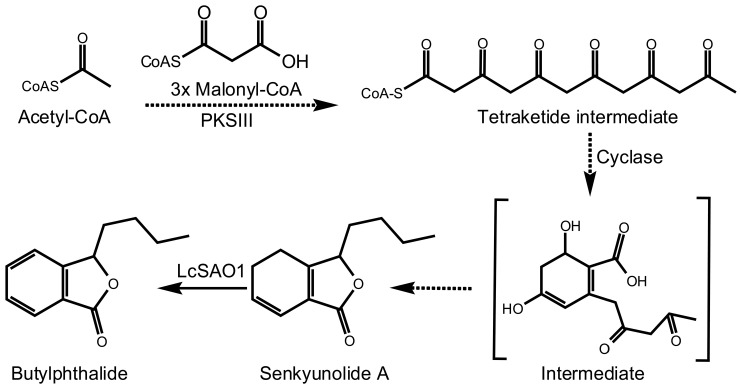
The proposed biosynthetic pathway of butylphthalide in *L. chuanxiong*. Proposed alternative butylphthalide biosynthetic pathways: A pathway involving one type III polyketide synthase (PKSIII) that catalyzes the condensation of acetyl-CoA and malonyl-CoA to form an unstable intermediate, followed by cyclase to form the scaffold of phthalide, then tailored by enzymes to form butylphthalide.

**Figure 2 ijms-24-17417-f002:**
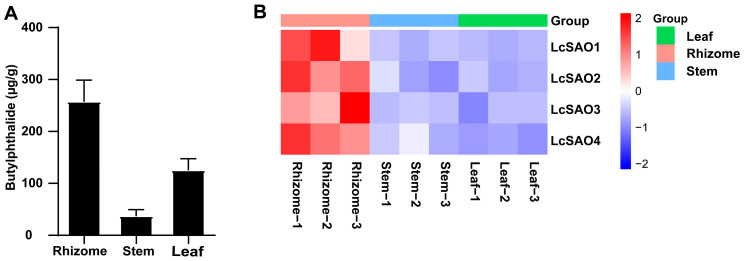
Screening the candidate genes. (**A**) Content of butylphthalide in three tissues of *L. chuanxiong* measured by UPLC-MS. The error bar denotes the standard error. (**B**) The expressional levels (fragments per kilobase of exon model per million mapped fragments) of 4 2-OGD genes in the leaf (L), rhizome (R), and stem (S) of *L. chuanxiong*.

**Figure 3 ijms-24-17417-f003:**
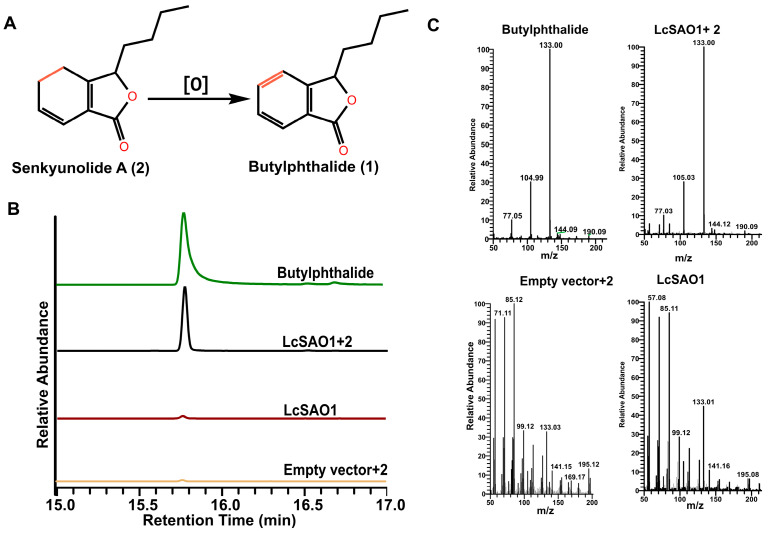
Functional characterization of the desaturation enzyme LcSAO1. (**A**) Proposed enzymatic reaction from senkyunolide A (**2**) to butylphthalide (**1**). (**B**) GC-MS analysis of 1 (*m*/*z* 133 to 77) in *N. benthamiana* leaves infiltrated with **2** or **1** to show the specific activity of LcSAO1. (**C**) GC-MS spectra of **1** in the total ion chromatogram mode.

**Figure 4 ijms-24-17417-f004:**
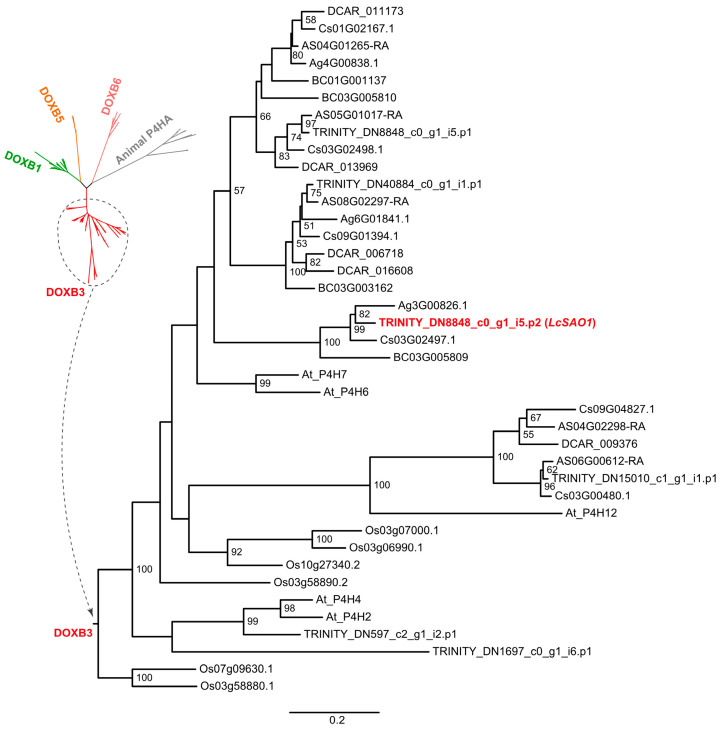
Phylogenetic tree of LcSAO1 and other DOXBs. The accession numbers of sequences used in this study are shown in Appendix A. P4H, proyl 4-hydroxylase.

**Figure 5 ijms-24-17417-f005:**
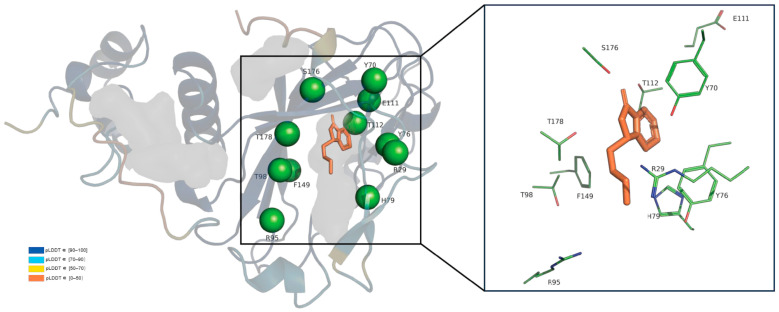
The complex model of LcSAO1 and senkyunolide. The cartoon of LcSAO1 was multi-colored, and the color codes correspond to pLDDT levels. Senkyunolide A was shown as magenta sticks. Candidate mutation sites in the active pocket are shown in green lines.

**Figure 6 ijms-24-17417-f006:**
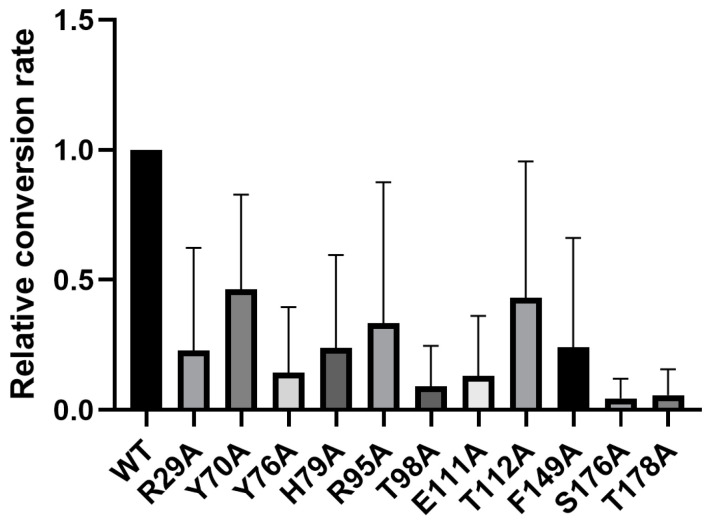
The relative conversion rates of wild-type (WT) LcSAO1 and mutants. In vivo assays of LcSAO1 and its variants using senkyunolide A as the substrate. Error bars indicate standard errors of three independent replicates.

## Data Availability

The datasets presented in this study can be found in online repositories. The raw RNA sequencing reads have been submitted to NGDC (CRA011442) and can be found at: https://ngdc.cncb.ac.cn/search/?dbId=gsa&q=CRA011442 (accessed on 1 November 2023). The other data underlying this article will be shared on reasonable request to the corresponding author.

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
