# Peer review of "LcSAO1, an Unconventional DOXB Clade 2OGD Enzyme from Ligusticum chuanxiong Catalyzes the Biosynthesis of Plant-Derived Natural Medicine Butylphthalide"

_ijms, 2023, doi:10.3390/ijms242417417_

Round 1
Reviewer 1 Report
Comments and Suggestions for Authors
The text below, contains comments on manuscript entitled “LcSAO1, an unconventional DOXB clade 2OGD enzyme from Ligusticum chuanxiong catalyzes the biosynthesis of plant-derived natural medicine butylphthalide”.
The manuscript is focused to elucidate the in vivo pathway for butylphthalide biosynthesis in various tissues of Ligusticum chuanxiong.
The manuscript is well written on sufficiently good English and grammar. The experimental section is properly designed. The obtained results fully explain the aim of the study.
I have some minor suggestions for corrections, listed:
Please formulate pore precisely the aim of your study, what is the reason, the novelty and the expected results. Introduction section is not the place to summarize your results.
To my opinion, the material and method section is too detailed in some points and could be shortened if possible.
I also think that a strong conclusion sentences, including future perspectives are necessary to summarize the preformed study.
Comments on the Quality of English LanguageMinor editing of English language required
Reviewer 2 Report
Comments and Suggestions for Authors
The manuscript submitted by Chen et al. reported that LcSAO1 catalyzed the desaturation from Senkyunoide A to Butylphthalide. My comments are shown as below. I hope that my comments are useful for the improvement of the paper.
Major points
1. I think the experiment using the recombinant protein expressed in E. coli have some problems as indicated below. Authors described that “Unfortunately, the enzyme reaction product was not detected by GC-MS (Figure S2C)” in lines 151-152. Figure S2C shows that Butylphthalide was detected in LcSAO1 and empty vector. Authors should explain the reason why product was detected in “empty vector”. Authors also should explain what is “empty vector” in Materials and Methods. Author described “Experimental validation through transient expression assays in Nicotiana benthamiana and recombinant E.coli expression systems corroborates thus transformative enzymatic activity” in Abstract (Lines 25-27). Your results using the recombinant protein expressed in E. coli didn’t support the LcSAO1 activity.
2. Chen et al emphasize that LcSAO1 catalyzes desaturation reaction. I think it is very likely thatthe hydroxylation occurs first, followed by aromatization to butyl phthalide. I highly recommend to check whether the hydroxylated compound is detected by the introduction of LcSAO1 to N. benthaminama.
Minor points
1. Make sure you italicize the scientific name.
2. Compound number should be written as bold font.
3. Line 145 m/z should be written as italics.
Comments on the Quality of English LanguageThe paper is well-written, but the explanation of the results using recombinant enzymes needs significant revision as described above.
Round 2
Reviewer 2 Report
Comments and Suggestions for Authors
It has been appropriately revised on the points I commented on.
Minor points
Line 97 “L. Chuanxiong” should be written as “L. chuanxiong”
Line 105 “Chuanxiong” should be written as “chuanxiong”
Line 179-180 Arabidopsis thaliana and Oryza sativa should be written as italics.